

# The potential of taxation records as a data source for historical climatology

Rudolf Brázdil[1,2], Jan Lhoták[1,3], Kateřina Chromá[2], Laurent Litzenburger[4]

[1]Institute of Geography, Masaryk University, Brno, Czech Republic
[2]Global Change Research Institute, Czech Academy of Sciences, Brno, Czech Republic
[3]Department of Historical Sciences, University of West Bohemia in Pilsen, Czech Republic
[4]Université of Lorraine, CRULH, Nancy, France

*Correspondence to*: Rudolf Brázdil (brazdil@sci.muni.cz)

**Abstract.** The tax liability of peasants significantly influenced their lives and the total monetary income of the country. The damaging effects of weather extremes on crop yields were considered grounds for tax relief. Administrative documentation connected with requests for tax relief can serve as an important source of data for historical climatology, as demonstrated by the example of the Prácheň Region in southwestern Bohemia during the 17th–19th centuries. Based on the first land registry system, only hailstorm damage to crops and fires qualified peasants for tax relief from 1655 CE, while the subsequent land registry system from 1748 CE extended this to include water damage from 1775 CE. Taxation data made it possible to analyse the spatiotemporal variability of significant hailstorms, water torrents, and lightning-caused fires, together with their impacts on agriculture and the lives of peasants during the 1655–1707 and 1748–1827 CE periods in the Prácheň Region, for which summary data at governmental and regional levels were preserved. Data related to weather damage were further supplemented with other documentary sources to create a chronology of significant hailstorms, water torrents, and additional weather extremes for the analyzed region. This study and its results clearly demonstrate the potential of taxation records – available in the Czech Lands as well as in many other countries – for historical-climatological research into past damaging weather events and their human impacts.

## 1 Introduction

Historical climatology, as a science that spans both climatology and, to a lesser extent, environmental history, uses a broad range of documentary evidence to study climate variability and its human impacts and responses over past decades and centuries (Brázdil et al., 2005, 2010; Carey, 2012; White et al., 2018, 2023). Institutional documentary evidence represents an important part of valuable datasets in cases where human activities documented by governmental, economic, or other institutions were influenced in any way by the weather and climate. Because obtaining information about weather and climate was not the primary aim of these sources, it must be secondarily extracted from institutional records, often using various statistical methods. Many historical-climatological studies have used and analyzed such sources, represented, for example, by data on the beginning of grain or grape harvests (e.g., Kiss et al., 2011; Wetter and Pfister, 2011; Možný et al.,



2012, 2016a, 2016b; Labbé et al., 2019), including wine must quality (Pfister et al., 2024), maritime or river transport (e.g., Tarand and Nordli, 2001; Leijonhufvud et al., 2010), rogation ceremonies (e.g., Martín-Vide and Barriendos Vallvé, 1995; Barriendos, 1997; Piervitali and Colacino, 2001; Domínguez-Castro et al., 2008; Tejedor et al., 2019), harvests and grain prices (e.g., Bauernfeind and Woitek, 1999; Esper et al., 2017; Pribyl, 2017; Ljungqvist et al., 2022; Brázdil et al., 2024;

Ljungqvist and Seim, 2024), public granaries (e.g., Collet, 2010; Brázdil et al., 2025b), famines, or mortality (e.g., Collet and Schuh, 2018; Ljungqvist et al., 2024), etc. Very little attention has been paid, however, to another available institutional data source: namely, taxation – which given its relationship to weather and climate effects could also be used to study past climatic patterns and corresponding human impacts and responses (e.g., Grove and Battagel, 1983; García et al., 2003; Huhtamaa et al., 2022; Gjerde et al., 2023).

The Czech Lands (now the Czech Republic, and hereafter CR) belonged to the regions where the existing taxation system allowed peasants affected by damaging weather events to request a tax reduction or exemption. Related rich taxation documents have been used in various types of historical-climatological studies. Koutný (1908), for example, published data from the Imperial Royal Land Financial Management in Brno concerning damaging hailstorms and corresponding tax alleviations for individual municipalities in Moravia (the eastern part of the CR) for 1896–1906 CE, which were later

presented in the form of maps (Brázdil et al., 2006). Matušíková (1999) used taxation records from the second half of the 17th century to analyse disastrous events in the Polabí region (in Bohemia, the western part of the CR). Particular attention to taxation documents related to tax relief for farmers affected by extreme weather events was given to South Moravia. Taxation data were used to analyse the occurrence of hydrometeorological extremes and their impacts, either in a single estate/domain (Brázdil and Valášek, 2003; Zahradníček, 2006; Chromá, 2011) or for multiple estates/domains together

(Brázdil et al., 2003, 2012; Dolák et al., 2013, 2015; Dolák, 2016). Extremes identified from taxation records were further used to compile long-term South Moravian series of floods (Brázdil et al., 2014) and hailstorms (Brázdil et al., 2016).

While the cited Czech studies focused mainly on the basic level of taxation data for individual settlements and estates/domains, particularly in South Moravia, the recent study shows how such data were handled and summarized at the governmental level in Bohemia, and how it is possible to move in the opposite direction – from the highest level of the

*Gubernium* (the government in Bohemia) to the basic level of individual domains or settlements. Moreover, while previous studies generally omitted the value of damage caused, the recent study also demonstrates the severity of weather events evaluated in terms of damage magnitude. Tax documentation serves as an interface between the consequences of weather extremes from a grassroots perspective in rural communities and changes in political responses to social demands during modern state formation. To demonstrate the potential of tax alleviation data for historical-climatological research, the results

of this analysis are presented using the example of the former Prácheň Region in southwestern Bohemia for two available time intervals during the 1655–1827 CE period.



## 2 Data

### 2.1 Region studied

The Prácheň Region, located in southwestern Bohemia, represented an historical administrative region of the Czech
Kingdom within the Austrian Habsburg monarchy (Fig. 1). It was named after Prácheň Castle near Horažďovice (for the
location of individual places, see Fig. A1), which was originally the administrative centre of the region (Sedláček, 1926).
The map of the Prácheň Region reconstructing the situation from 1654 CE and the historical map from 1840 CE are shown
in Fig. 1. This region was administratively divided into domains, which changed in number and area between the second half
of the 17th century and the first half of the 19th century due to temporary connections under the same ownership. For
example, Palacký (1848) mentioned a total of 79 domains as of 1846 CE, and the total area – nearly corresponding to the
catchment of the Otava River – represented 4 580.45 km², accounting for 8.82 % of the entire territory of Bohemia
(Schnabel, 1848).



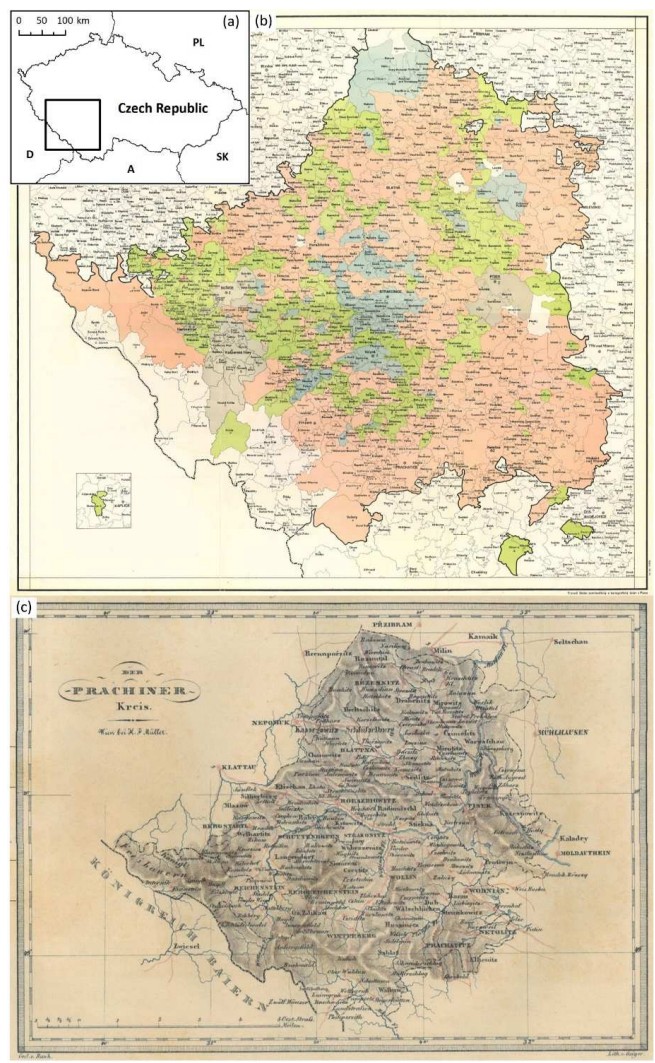

**Figure 1. The Prácheň Region in southwestern Bohemia shown (a) within the territory of the Czech Republic (A – Austria, D – Germany, PL – Poland, SK – Slovakia), (b) on a map reconstructing the situation from 1654 CE (Haas, 1954), and (c) on a historical map from 1840 CE (Schmidl, 1840).**




### 2.2 Taxation system and data

The first land registry in Bohemia, the Roll of Assessment (*Berní rula*), was compiled in 1653–1655 CE and later revised in 1667–1682 CE. The first instruction for compensating damage caused by fires or hailstorms to peasants in Bohemia was sent to all regional offices on 18 June 1655. Compensation was based on tax relief – three years for fire damage and one year for hailstorm damage. The extent of damage was expressed in terms of taxation units called *osedlý*, which represented an ideal peasant farm in Bohemia and varied across regions depending on soil fertility. As for the tax relief process (Fig. 2), an affected domain informed the regional administrator about the damage, who then prepared a damage report that was sent to the land governor. The governor evaluated all reports (Fig. 3a) and responded with a decision to the regional administrator, who then took it into account during tax allocation. Information on tax relief was recorded as notes on individual rolls of assessment. Due to various issues with damage reporting, the corresponding instruction was reissued on 2 December 1686, this time with a special form for hailstorm damage that included reporting for individual peasants and the proportion of damaged fields relative to all cultivated land with crops. The difficulties inherent in this specific declaratory system reflect broader problems of the tax system, including attempts at tax evasion by the nobility and discrepancies in calculations between the crown countries and the Court of Vienna. These issues motivated major tax reforms in the first half of the 18th century and the development of a land register (Klinger, 2007).

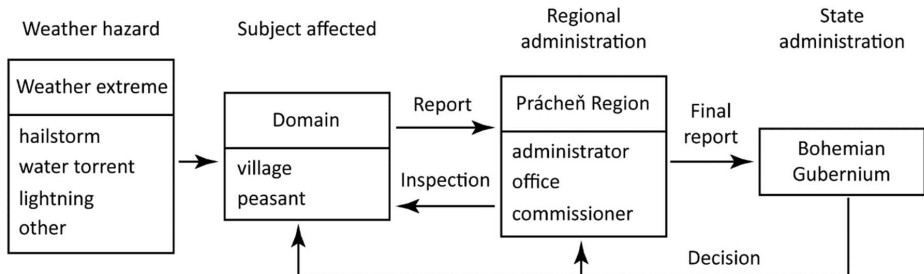

**Figure 2. A diagram of the administrative process connected with tax relief for peasants in the case of weather damage.**

The tax alleviation system for weather damages to peasants changed significantly after the declaration of the Land Registry of Maria Theresa (*tereziánský katastr*) on 1 May 1748 CE, which came into effect on 1 November. The old system of compensation based on *osedlý* was replaced by cash payments. Instead of tax relief, a new disaster fund was created, from which affected peasants could be paid in money (in *zlatý* – gulden), with an annual allocation of 170 000 gulden (Pekař, 1932). To express the value of the gulden in 1779 CE from a grassroots human perspective: for example, a bricklayer's journeyman in Vienna earned a daily wage of 0.4 gulden (Pribram, 1938), and in Prague, one gulden could buy a daily portion of rye bread (0.95 kg) for 21 days (Honc, 1977). In 1754 CE, taxes in Bohemia brought in 3 180 480 gulden from rustical land (i.e., land held by peasants), while the total tax revenue was 6 076 093 gulden (Pekař, 1932; Klinger, 2007).



Thus, the mentioned direct aid of 170 000 gulden represented 5.3 % of rustical and 2.8 % of total tax income. Despite yearly variations in Bohemia's tax contributions, these figures indicate that the volume of direct aid or tax relief granted by the state

during this period was relatively easy to absorb from a budgetary perspective, while representing a significant response from the viewpoint of affected communities.

The edict from 6 September 1748 defined a procedure for reporting damage caused by fire or hailstorms, requiring that an affected domain immediately inform the corresponding regional office – specifically its regional commissioner – who was then obliged to inspect the situation *in situ* and prepare a related report. These reports (Fig. 3b), which were subsequently

submitted to the *Bohemian Gubernium* for a final decision, have survived with varying levels of detail for both types of damage after 1748 CE. Awarded money was paid directly to peasants in guldens (with some time delay). For example, in 1768 CE, a total of 123 039 guldens was paid to compensate for hailstorm damage across the whole of Bohemia (archival source AS13).

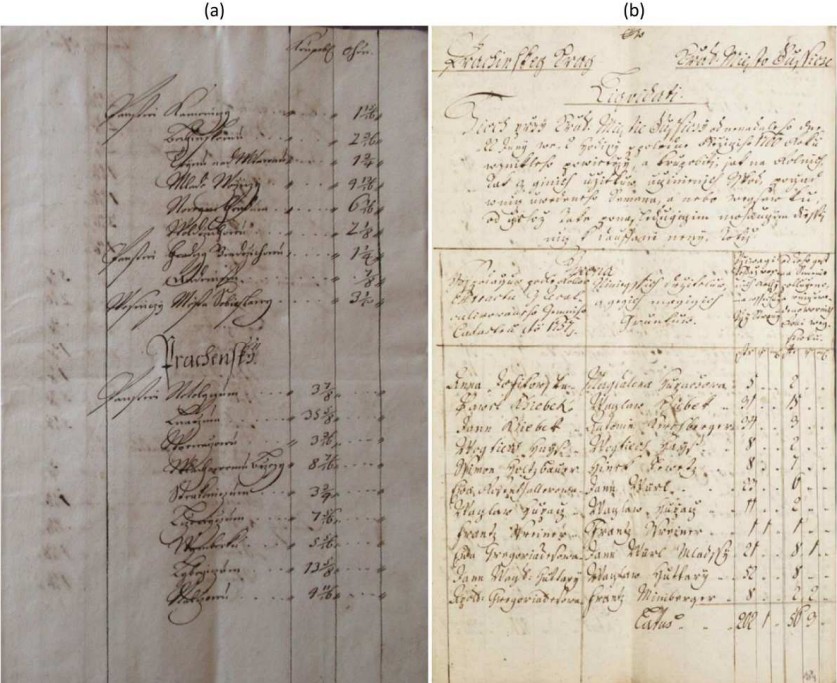

**Figure 3. Examples of tax relief records due to weather damage: (a) concept of damage assessment from 1688 CE (AS10), (b) liquidation reports from 1760 CE (AS14).**

Hailstorm damage to fields was expressed in *strych* (1 *strych* = 0.285 ha), which quantified the extent of the damaged area of fields, gardens, and vineyards – damaged by one third, one half, or in full. This classification then determined the conversion



to a specific tax relief in guldens. From 1775 CE, the reporting of key events for tax alleviation was extended from fire and
hailstorm to also include water damage. Other types of damage (e.g., frosts, strong winds, etc.) were not compensated.
Hailstorm damage was compensated at twice the rate of water damage, which also considered the flooding of meadows. Due
to the non-uniform reporting of the aforementioned damages, the decree issued on 26 August 1780 CE introduced new
instructions on how to handle reported damages, further specified in a circular for the regional offices (*Circulare in alle
Kreise, die Feuer-, Wetter-, und Wasserschaden-Angelegenheiten betreffend*) dated 22 October 1780. The circular defined a
reporting scheme for the related damages. In the case of hailstorm damage, it included the names of peasants, the names of
damaged fields, total annual tax, the area with winter and spring crops, and the portion of these areas that was damaged
according to the three thresholds (1/3, 1/2, and full). Hailstorm damage was accepted only for events that occurred generally
from the beginning of May and from mid-June in mountain regions. The scheme for water damage, instead of reporting
bruised areas from hail, included the area of fields (in *strych*) and meadows (in wagons of hay) clogged by sand and stones
or totally destroyed, and finally the total damage expressed in guldens. A special printed instruction (*Instruction, nach
welcher der Wirtschaftsbeamte, und Contributionsvorsteher die vorfallende Feuer-, Wetter-, und Wasserschaden im
Königreiche Böhmen aufzunehmen und zu liquidiren haben wird*) then defined the obligations of responsible officers at the
regional and governmental levels on how to proceed with the adjustment of reported damage.

The Land Registry of Maria Theresa remained valid until 1789, when it was temporarily replaced for two years by the Land
Registry of Joseph II (*josefinský katastr*). Subsequently, both land registers were used in combination. The stabilization of
the state economy after the Napoleonic Wars (1803–1815) was reflected in a decree of the Bohemian *Gubernium* dated 23
March 1819, which allowed tax compensation for those who were unable to pay taxes duly due to disaster damage. This
compensation was extended from peasant land (rustical) to also include landlord land (dominical). Ultimately, tax relief was
preferred over direct aid as a means of supporting communities. Recorded damage was entered into special forms. The basic
unit for land registers was the tax unit, of which there were more than the number of domains in the Prácheň Region, since
several tax units could be owned by a single owner. Specifically, in 1655 CE there were 165 tax units (Haas, 1954), 148 in
1756 (Burdová et al., 1970), 141 in 1827 (AS11), and 140 in 1846 (AS12).

The records were excerpted from official documentation at both the regional and pan-Bohemian levels to cover the entire
analysed territory. Concepts of notices from the *Gubernium* tax office in Prague, directed to the regional council president of
the Prácheň Region, have survived for the years 1653–1707 CE (AS9). Except for a few cases, these notices did not include
daily but only annual data on damaging events (as they were responses to announcements previously sent by regional council
presidents), expressed in the taxation unit *osedlý*.

The new system of compensation from 1748 CE (AS1, AS2, AS9) was represented by original damage adjustments that the
*Gubernium* used for the assessment of tax relief. These documents contained detailed information on the date of damage, its
magnitude, and sworn statements from participants in the local investigation. These materials have survived (with some
gaps) for the third quarter of the 18th century for the entire analysed region in the registry of the Prácheň Region (AS14).
Starting from the second half of the 1780s, only the concepts of reports from the central institution (*Gubernium*), sent to



regional offices and containing proposals for damage compensation, were again available. Depending on the personality of the corresponding scribe, the exact date of the damaging event was sometimes mentioned, and documents showing the actual

liquidation investigation were also irregularly attached (AS3, AS4, AS5, AS6). However, these records have not survived completely, as no related documents were found for the period 1806–1813.

### 2.3 Documentary data

To complement and compare damaging weather phenomena extracted from taxation data, similar information was collected from the historical-climatological database of the Institute of Geography, Masaryk University in Brno, as well as from other

documentary data extracted from various narrative sources in the Prácheň Region. This concerned, in particular, convective storms, with which further investigated weather phenomena – such as hailstorms, torrential rains with water torrents, and lightning strikes – were associated due to their damaging effects on agriculture and property.

### 2.4 Climatic data

To document the general climatic background of the entire analysed period 1655–1827 CE, two climate reconstructions were

used:

(i) the temperature reconstruction of Central Europe by Dobrovolný et al. (2010), which compiles documentary-based temperature indices from Germany, Switzerland, and the Czech Lands (1500–1854 CE), and mean temperatures from 11 Central European stations (from 1760 CE onwards), representing very well the patterns of the CR (see Brázdil et al., 2022);

(ii) the precipitation reconstruction of the CR by Dobrovolný et al. (2015), which combines documentary-based Czech

precipitation indices (1501–1854 CE) and mean areal precipitation totals (from 1804 CE onwards).

### 3 Methods

Taxation data extracted and critically evaluated from sources described in Sect. 2.2 were used to create a database, which included the following information: date of the damaging event, type of the event, affected domain and villages, tax relief accepted by the *Gubernium*, estimated damage, remarks on the event, and archival source. This database was used for further

analyses. Of the total 2 134 collected records from the period 1655 to 1827 CE, 1 027 (48.1 %) concerned only damages caused by fires unrelated to weather, and were therefore not considered further in this study.

Descriptions of damaging weather events in the taxation records used a broad range of Czech and German terminology:

(i) hailstorm damage: *Elementarschaden, Hagelwetter, Schauerwetter, Schlossen, Wetter, Wetterschlag, Wetterbeschädigung*, and *Wetterschaden* in German; *krupobití* and *povětří* in Czech;

(ii) water damage: *Wasserfluth, Wasserguß*, and *Wasserschaden* in German; *příval, liják*, and *průtrž mračen* in Czech;

(iii) lightning damage: *Donnerschlag, Donnerstrich*, and *Donnerwetter* in German; *bouřka, hromobití, hromové udeření*, and *úder blesku* in Czech;





(iv) other weather damage: *vymrznutí žita* (freezing of rye after winter), *sucho* (drought), and *vítr* (wind) in Czech.

The extracted taxation data in the Prácheň Region generally cover two basic periods, namely 1655–1707 and 1748–1827 CE,
with a data gap between 1806 and 1813. Tax relief was expressed explicitly in taxation units (*osedlý*) during the first period,
while in the second period it was expressed in *zlatý* (gulden), *strych* (for fields), and in the number of wagons of hay for
meadows, which were used by peasants as grazing land and as the main source of hay for livestock.

Information from the above database was used to analyse annual fluctuations in the frequency of damaging events due to
hailstorms, water, and fires caused by lightning, the number of affected domains, and the magnitude of damage (awarded or
estimated).

## 4 Results

The following analysis is based on 303 individual taxation records for the period 1655–1707 CE and 804 such records for the
years 1748–1827 CE (with missing data for 1806–1813). This represents an average of 5.7 damage records per year in the
first period and nearly double, 11.2 records per year, in the second period. Results were structured according to the four
groups (i)–(iv) of individual damaging weather events as specified in Sect. 3 and further summarised, complemented by
weather data from other documentary evidence from the Prácheň Region.

### 4.1 Individual weather-related damages

### 4.1.1 Hailstorm damage

Hailstorm was the only damaging weather event recorded continuously throughout the entire analysed period. The first
hailstorm damage was mentioned for the Štěkeň domain (municipalities Domanice, Černíkov, and Droužetice) on 23 June
1655, when "*winter and spring crops were bruised by hail*," but without an expression of the corresponding damage (AS8).
Hailstorm damage was detected exclusively for 32 years during the 1655–1707 CE period (Fig. 4), but only the first cited
event from 1655 was exactly dated in the taxation records. According to the number of affected domains, the maximum of
30 domains was reported for 1688, followed by 24 domains in 1707 and 20 domains in 1693. The highest tax relief of 170.9
*osedlý* was also mentioned for 1693, followed by 145.5 *osedlý* at 16 affected domains in 1669. Nearly identical tax relief was
recorded in 1688 and 1707 – 107.8 and 107.6 *osedlý*, respectively.





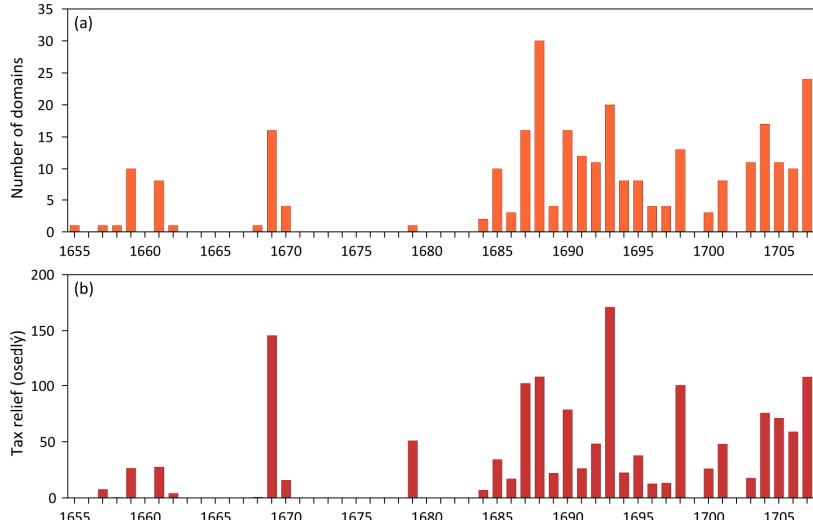

**Figure 4. Hailstorm damage in the Prácheň Region during the 1655–1707 CE period: (a) total annual number of affected domains, (b) tax relief expressed in multiples of *osedlý*.**

In the following 1748–1827 CE period, hailstorm damage was identified in 35 years. The related chronology showed year-by-year occurrences in several time intervals: 1756–1760, 1762–1765, 1789–1797 and 1820–1827 CE (Fig. 5). The dominantly highest number of 43 affected domains was detected in 1796, being followed by 31 domains in 1779 and 28 in 1760, while hailstorm damage in several other years concerned only a few domains or was even missing. Probably because of incomplete data of tax relief, cited three years did not appear among those with the highest awarded compensation: tax

relief of 7 327 guldens was reported for 24 domains in 1763, 5 267 guldens for 18 domains in 1774 and 4 515 guldens for 17 domains in 1756. No data about damage survived for 1820–1827. While hailstorm damage without exact date specification was mentioned for 10 years, the dominant occurrence of hailstorm damage in those with exact dating was identified for June (48.7 %), followed by July (20.3 %), August (17.7 %), May (11.5 %) and September (1.8 %).



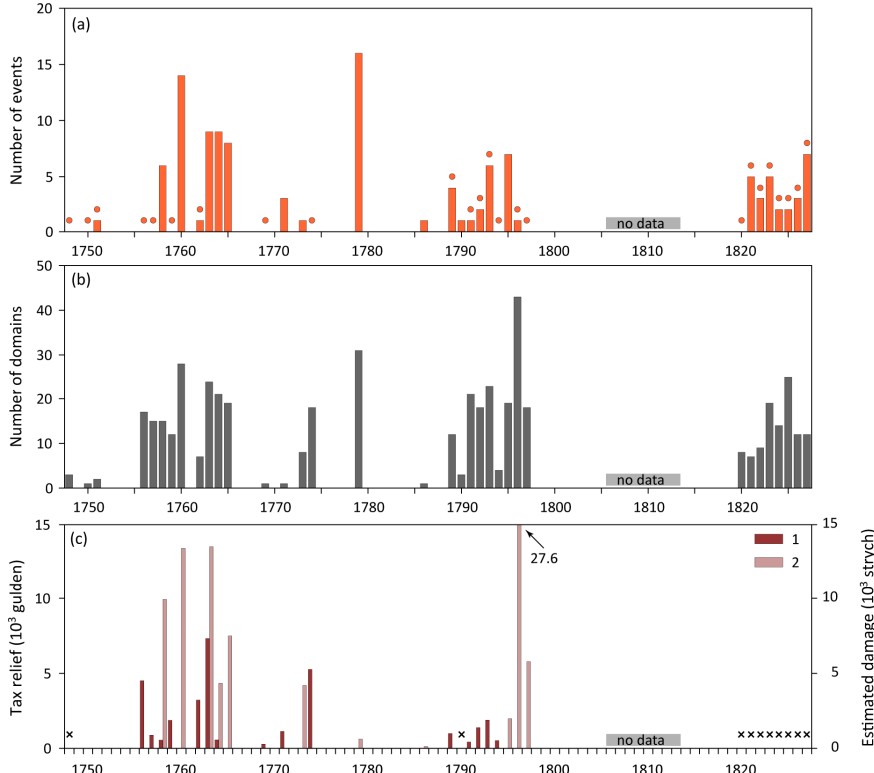

Figure 5. Hailstorm damage in the Prácheň Region during the 1748–1827 CE period: (a) the number of dated hailstorm damage events (circle indicates non-dated events), (b) total number of affected domains, (c) awarded tax relief expressed in guldens (1) and estimated damage in *strych* (2); x indicates no tax relief/damage data.

### 4.1.2 Water damage

Tax relief for water damage began to appear in the second time interval, 1748–1827 CE. Water damage was associated with torrential rains (downpours) and the resulting surface water torrents that carried stones and gravel onto fields and meadows, while also causing soil erosion on arable land. Damaging effects were often intensified by the simultaneous occurrence of hailstorms. Although water damage was reported for the first time for the Vimperk domain in 1750 (AS7), and again as a water torrent (*příval*) together with a hailstorm at Hojsova Stráž on 13 July 1763 – affecting 144 *strych* (out of a total of 257 *strych*) with awarded tax relief of 60 guldens (AS13) – such records began to appear more frequently from 1789 CE onward, and were recorded in total for 22 years (Fig. 6). The highest numbers of domains affected by water damage were detected in the years 1779 (19), 1791 (20), 1793 (22), 1796 (22), and 1825 (17), with year-by-year damage reported for 1789–1797 and




1820–1827 CE. Among the limited available tax relief data, the highest recorded damage occurred in 1793 CE, with approximately 1 724 guldens awarded to 22 affected domains. The highest estimated water damage, represented by 2 898 *strych* of fields and 1 238 wagons of hay, concerned 22 domains in 1796 CE. From exactly dated cases of water damage

associated with torrential rains, the most frequent occurrences were in June (47.6 %), followed by August (20.6 %), July (14.3 %), May (12.7 %), and September (4.8 %). However, water damage was also reported on 2 December 1779 for Kučeř and Volary.

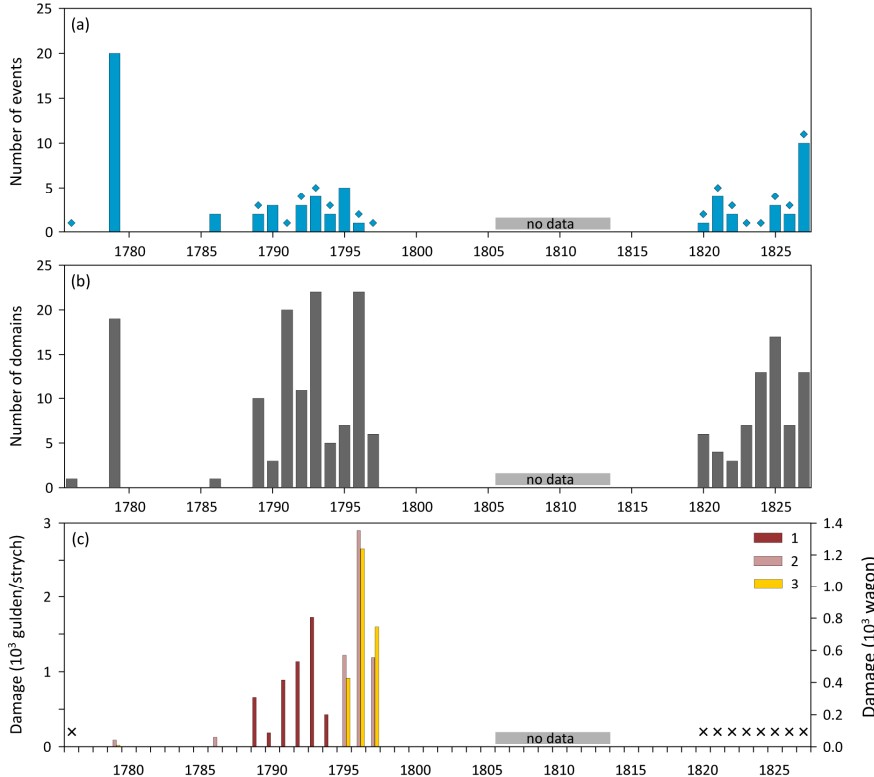

**Figure 6. Water damage in the Prácheň Region during the 1776–1827 CE period: (a) number of dated water damage events**
**(diamond indicates non-dated events), (b) total number of affected domains, (c) damage expressed as tax relief in guldens (1) and**
**as estimated damage in *strych* (2) or in number of hay wagons (3); x indicates no damage data.**

### 4.1.3 Lightning damage

Lightning strikes during thunderstorms sometimes caused fires in buildings or other farm structures. In the available taxation reports, the first fires due to lightning were recorded in 1758 CE, specifically on 10 August for the municipalities of Láz and





Nepodřice, and on 24 August for Věšín and Vranovice (AS13). Similar events were further mentioned in 1760, 1763, and
1764. After that, such reports did not occur until the 1790s, during which only 1791 and 1799 lacked note of any lightning-caused fires (Fig. 7). The next lightning-induced fire was reported in 1802, and after a long break, again in 1816, 1818, 1822,
1824, and 1827, i.e. across 18 years and for a total of 39 municipalities. On 21 July 1794, a lightning strike caused fires in
five municipalities. Reported fires caused by lightning occurred most frequently in August (32 %), followed by June (24 %),

July (20 %), May (12 %), and September (8 %), with a single case also reported on 2 October 1792 (4 %). It appears that the
damage estimated by the regional administration was always significantly higher than the tax relief awarded by the
*Gubernium*. For example, in the above-mentioned case of four municipalities affected by fire in 1758 CE, only 331 guldens
were awarded in total, although the estimated total damage amounted to 2 237 guldens (AS13).

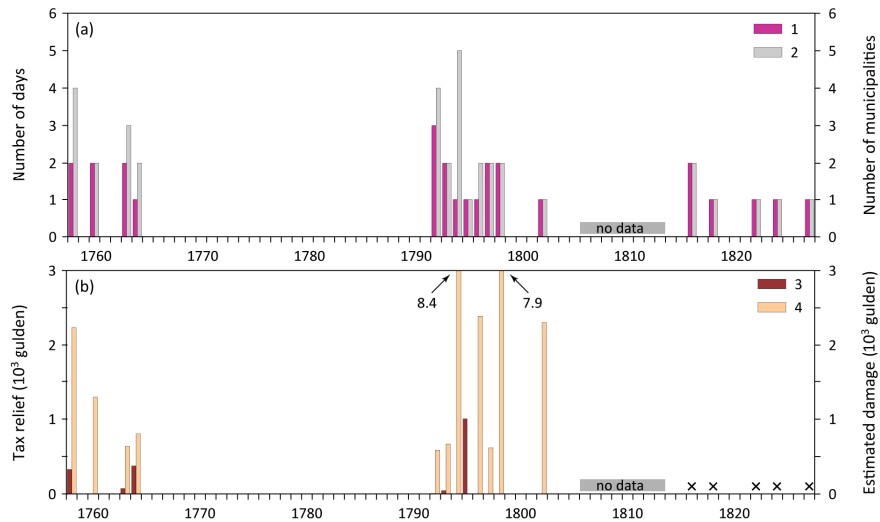


**Figure 7. Lightning damage (fires) in the Prácheň Region during the 1758–1827 CE period: (a) total number of days (1) and affected municipalities (2), (b) tax relief (3) and estimated damage (4) in guldens; x indicates no tax relief/damage data.**

### 4.1.4 Other weather damage

In addition to the three previously discussed weather phenomena, taxation records also reported several other types of
events. Freezing of winter crops, particularly rye, was first recorded in the Prácheň Region in 1751 CE (AS9). During the
winter of 1757/58, rye froze in the fields due to "*long lying large snow*" in the domains of Přečín and Dobrš (578 *strych*, 20
municipalities), Vimperk (2 089 *strych*, 11 municipalities), and Volyně (645 *strych*, 21 municipalities) (AS13). Additional
information on winter crop freezing appeared for 1767 in a table summarising related damage across individual regions of
Bohemia. The Prácheň Region was only slightly affected, with 99 *měřice* (19.0 ha) reported for wheat (2.4 % of all





Bohemia) and 2 957 *měřice* (567.7 ha) for rye (0.8 %) (AS1). The regional administrator also mentioned freezing of winter crops in his report from 26 April 1770 (AS2), but this event was not listed among those eligible for tax relief.

A note stating that "*no rain for long time, still cold in May and wind drying up fields more*" appeared in records for 14 domains in 1758 CE (AS13), but again, no tax support was granted for crop failures caused by drought. On 4 January 1814, a "*fire as consequence of sharp wind*" was reported at Malý Kozí Hřbet in the Kašperské Hory domain (AS5).

**4.2 Weather-related damage from tax records in 1655–1827 CE**

Summarising all weather-related damage recorded in taxation documents for the Prácheň Region over both analysed periods, the first period (1655–1707 CE) featured only hailstorm damage, which occurred in 60.4 % of all considered years (32 years) (see Fig. 4a, which indicates hailstorm years). In contrast, during the second period (1748–1827 CE, excluding the years 1806–1813 with no data), some form of weather damage was documented in more than half of the years – specifically, in 43

years (59.7 %) (Fig. 8). Within these 43 years, the proportions of individual types of weather damage were as follows: 81.4 % (35 years) for hailstorm damage, 51.2 % (22 years) for water damage, 41.9 % (18 years) for fire damage due to lightning strikes, and 11.6 % (5 years) for other weather damage. This indicates that annual taxation records often included more than one type of weather-related damage. For instance, simultaneous occurrences of hailstorm and water damage on the same domains were confirmed for 19 years during 1748–1827 CE. In 1791 CE, both damaging weather events were accompanied

by fire in eight domains, although it was not specified whether the fire was caused by lightning strikes.





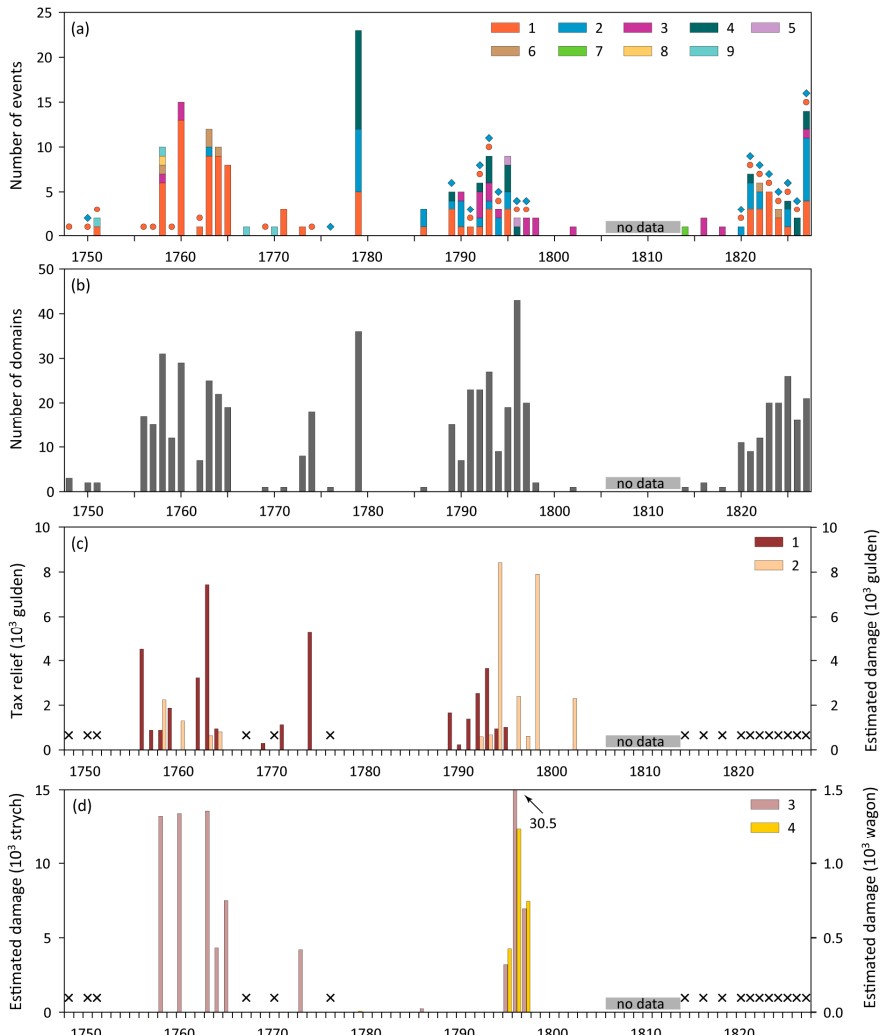

**Figure 8. Total annual weather-related damage in the Prácheň Region during the 1748–1827 CE period: (a) the number of reported damaging weather events (1 – hail, 2 – water torrent, 3 – lightning, 4 – hail and water torrent, 5 – hail, water torrent and lightning, 6 – hail and lightning, 7 – wind, 8 – drought, 9 – freezing; a circle indicates non-dated hail, a diamond indicates non-dated water torrent), (b) the total number of affected domains, (c) tax relief (1) and estimated damage (2) in guldens, (d) the estimated damage in *strych* (3) and in number of hay wagons (4); x indicates no tax relief/damage data.**

Because some tax records also report weather damage at the level of individual municipalities within a given domain, the spatial distribution of the four cases with the highest number of affected municipalities can be presented (Fig. 9):





(a) **15 June 1758**

Hailstorm damage on 15 June 1758 was reported for 26 municipalities, with a further 33 municipalities added due to damage reported for both 13 and 15 June, resulting in a total of 59 affected municipalities. The affected areas formed a main core in the northeastern part and a smaller cluster in the southeastern part of the region.

(b) **13 July 1763**

In addition to 26 municipalities with hailstorm damage on 13 July 1763, another 12 were reported as affected by damage recorded on this day as well as on 30 June, bringing the total to 38 municipalities. The affected municipalities were grouped into three clusters, with the largest in the southern part of the region. Only the municipality of Hojsova Stráž was damaged by both hail and water.

(c) **21 June 1765**

Taxation records reported hailstorm damage on 21 June 1765 for 42 communities and the Starosedlský Hrádek domain. Hailstorms associated with thunderstorm clouds occurred along a west–east line in the southern part of the region. On the same day, thunderstorms were also reported in Moravia. A downpour causing a flash flood heavily damaged houses, farm buildings, and roads, and drowned sheep, cows, and a horse in the Šumperk region (Polách and Gába, 1998), while a lightning strike around noon killed a woman in Přerov (Lapáček, 2003) (Fig. A1).

(d) **31 July 1779**

According to taxation records, 51 municipalities were affected by hailstorm damage, and three municipalities by both hailstorm and water damage on 31 July 1779. These municipalities were located along a line extending from the southwest to the northeast. This event represents the case with the highest daily number of affected municipalities documented in the Prácheň Region for the entire analysed period.




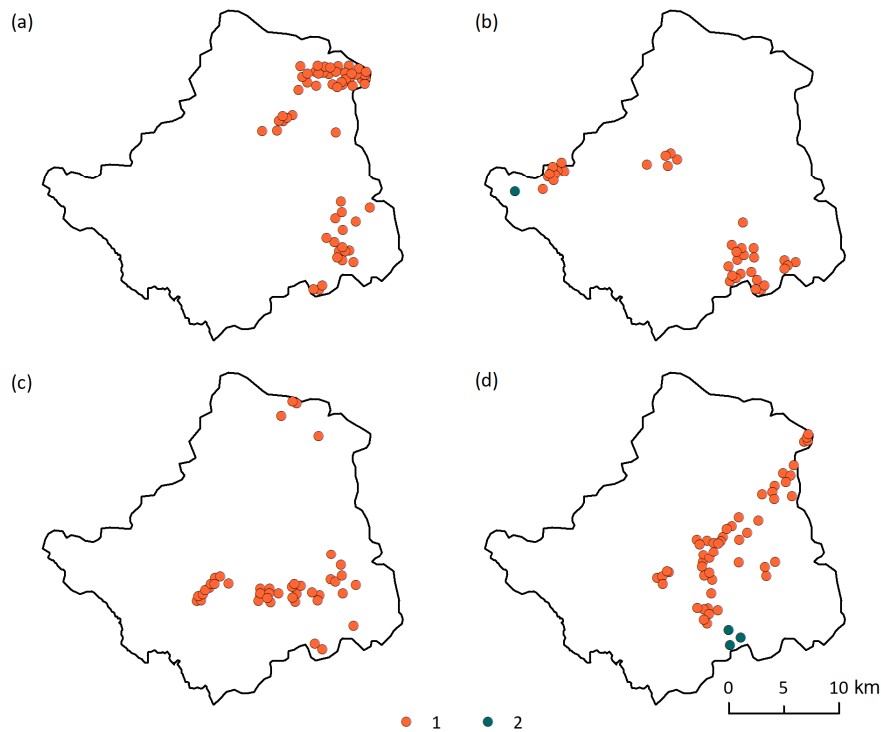

**Figure 9. Spatial extent of four outstanding damaging weather events with the highest number of affected municipalities: (a) 15 June 1758, (b) 13 July 1763, (c) 21 June 1765, (d) 31 July 1779; 1 – hail damage, 2 – hail and water torrent damage.**

**4.3 Summary of damaging weather events in 1655–1827 CE**

The annual numbers of damaging weather events in the Prácheň Region in Fig. 8a, derived from taxation records, can be complemented by other events with damaging impacts recorded in additional documentary evidence. This includes, for example, records from public granaries (Brázdil et al., 2025b), in which poor or failed grain harvests at various domains between 1789 and 1827 CE were attributed particularly to damaging hailstorms or "bad weather": 1797 – Hlavňovice; 1798 – Kolinec; 1802 – Velhartice; 1803 – Žichovice; 21 August 1806 – Žichovice; 16 August 1808 – Žichovice; 1812 – Žichovice (seven municipalities affected), Nalžovy; 1821 – Dlouhá Ves, Horažďovice. Great heat (probably with drought), leading to the failure of oats in 1796, was mentioned exclusively for the Albrechtice domain. Two municipalities of the Horažďovice domain recorded damage from torrential rain in 1822, as did a granary building of this domain in Třebomyslice, where 1 *měřice* of rye (61.5 L) and 5 *měřice* of oats (307.5 L) were irrecoverably damaged during the thunderstorms on 10 June 1827 (AS16).



But the chronology of damaging weather events can also be confronted and further complemented with information from published (see, e.g., Elleder et al., 2014, for floods) or archival documentary sources from the analysed region, as shown in
several of the following examples. In Netolice, hailstorm damage was recorded on 25 August 1659 and 13 July 1687, while torrential rain with a water torrent caused damage there on 10 July 1663 (Konzalová, 2013). Notes from a teacher in Němčice reported, among other events, damaging hailstorms on 13 July 1722 and 23 August 1740, water damage in 1731 and after a downpour on 30 June 1763, and hail accompanied by fires caused by lightning on 2 July 1730 (Starý, 2010). Reports from Volyně mentioned water damage to fields and meadows in spring 1775, and a water torrent that swept away
hay for livestock in 1782 (Teplý, 1933). Damage by hail and partly by water to cereal fields was recorded on 14 June 1796 in Sušice (AS17). Torrential rain and hail heavily damaged fields with crops on 23 May 1820 in Bukovník (AS18).

Based on the above tax relief data and cited documentary sources, an improved series of the annual numbers of damaging weather events in the Prácheň Region during the 1655–1827 CE period is shown in Fig. 10, taking into account the following events separately: hail, water torrent, lightning strike, thunderstorm, flood, windstorm, freezing, heat, and drought. Because
of the many cases of non-exactly dated hailstorm events during 1655–1707 and the generally lower number of other documentary records before the mid-18th century, the analysed period was split into 1655–1747 (Fig. 10a) and 1748–1827 (Fig. 10b). From tax relief data, a total of 218 damaging weather events were identified, in addition to 53 hail and 17 water torrent events reported without exact dating (only by year) and likely including many more individual events. This dataset was further complemented by another 148 events from documentary sources in the studied Prácheň Region, partly covering
years not represented in the tax relief data and partly slightly increasing their annual numbers. Besides the dominant year 1779 CE with 35 recorded events (Fig. 10b), higher numbers were recorded in 1758–1765, 1789–1797, and 1820–1827. On the other hand, during 1655–1827 CE, no damaging weather events were found for 49 years (28.3 %) in all available documentary evidence.



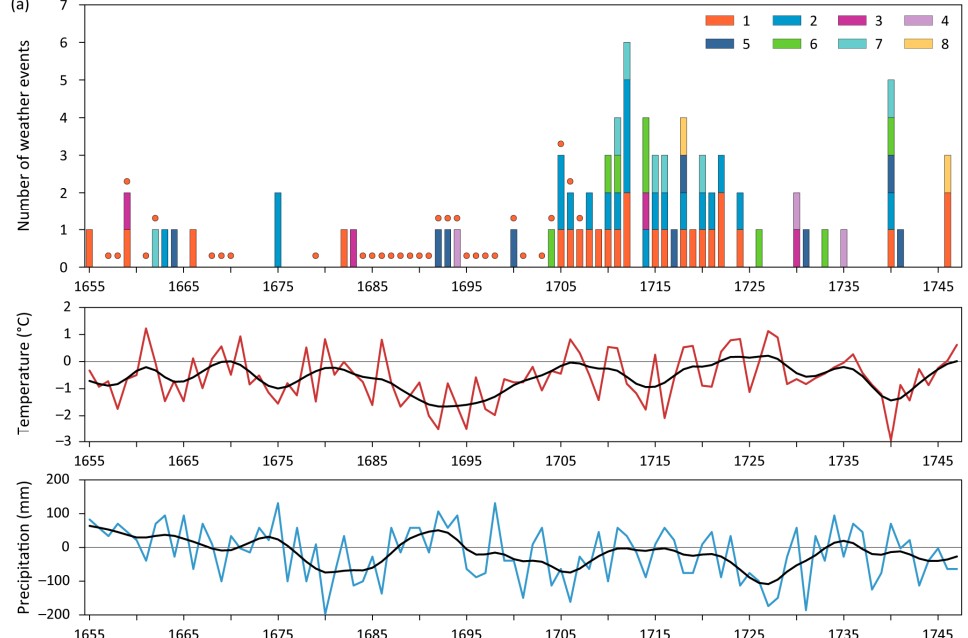

**Figure 10. Fluctuations in annual numbers of damaging weather events (1 – hail, 2 – water torrent, 3 – lightning, 4 – thunderstorm, 5 – flood, 6 – wind, 7 – freezing, 8 – drought, 9 – heat; a circle indicates non-dated hail, a diamond indicates non-dated water torrent) based on tax relief data and other documentary sources in the Prácheň Region in comparison with mean annual temperatures and precipitation totals in the Czech Lands during (a) 1655–1747 and (b) 1748–1827 CE. Temperature and precipitation data (Dobrovolný et al., 2010, 2015) are expressed as deviations from the 1961–1990 reference and smoothed by a 10-year Gaussian filter.**



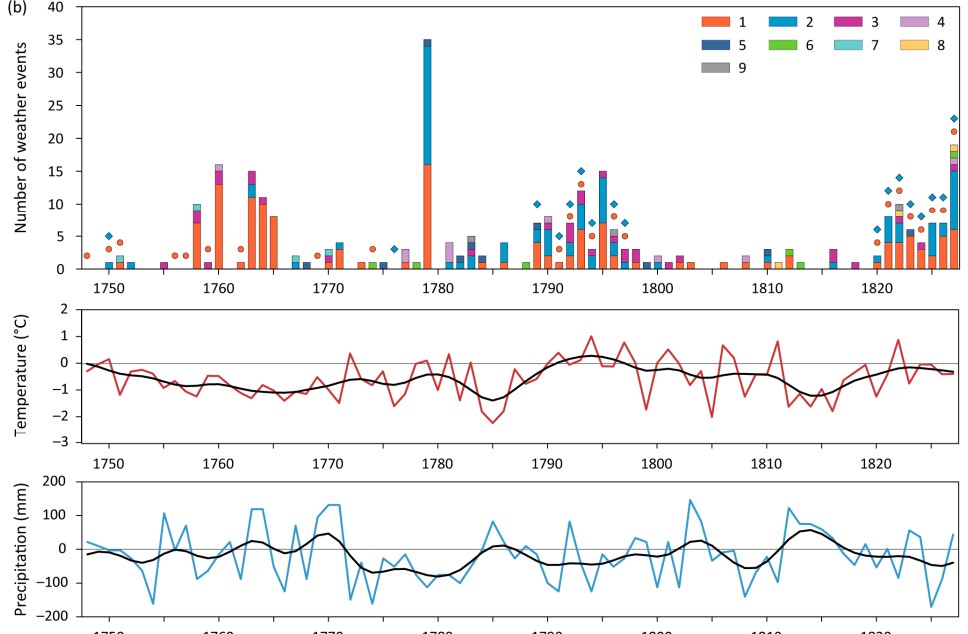

**Figure 10 – continued.**

To demonstrate the general climatic background of the analysed period, Fig. 10 also shows fluctuations in mean annual temperatures and precipitation totals in the Czech Lands (Dobrovolný et al., 2010, 2015), expressed as deviations from the

1961–1990 reference. Smoothed annual temperatures were generally below the reference mean throughout the entire period, except for a few short intervals with values close to or slightly above the mean. In the case of annual precipitation, smoothed totals fluctuated around the reference mean, showing both relatively wetter and drier periods during 1655–1827. However, due to the high temporal inhomogeneity in the series of damaging weather events, it is impossible to identify any clear relationship between their annual frequency and specific temperature or precipitation patterns.

**5 Discussion**

**5.1 Data uncertainty**

Using documentary evidence in historical-climatological research is strongly limited by the availability of corresponding data sources, as pointed out in several studies (e.g., Brázdil et al., 2005, 2010; White et al., 2018). This also applies to institutional evidence represented in this study by taxation records, which is particularly reflected in data gaps between 1708



and 1747 CE and further in the years 1806–1813 CE, as well as in the unavailability of records after 1827 CE.
Corresponding data may have been lost in subsequent years or discarded when they lost their importance for the relevant
institution, i.e. after the *Gubernium*'s decision regarding the approval or refusal of the requested tax relief.

Besides missing taxation data for the analysed time intervals, other data uncertainties must also be taken into account in the
interpretation of the study results. Another limiting factor until 1775 CE was that governmental compensation existed only

for damage caused by hailstorms or fires (see Sect. 2.2), which thus excluded the recording of other weather extremes.
Outside of this framework, disastrous events were recorded only by the nobility; however, the quality of such records
depended on the individual approach of the noble estate and was limited by the borders of the corresponding domain (*cf.*
Matušíková, 1999). Because some of the mentioned events were dated only by year or month, it is difficult to determine the
frequency of related events within a particular year. This can only partly be supplemented by reports from other available

documentary evidence. Concerning the damage caused, it is difficult to compare it over longer time intervals due to changing
expressions of damage, as specified in Sects. 2.2 and 3. Moreover, the timing of a hailstorm or water torrent was important,
as damage occurring before the harvest resulted in greater yield losses. This is evidenced by the rule that hailstorm damage
was accepted for tax relief only from the beginning of May (from mid-June in mountain regions). From a meteorological
point of view, the spotty character of hailstorm or torrential rain occurrence, connected with the development of individual

Cumulonimbus clouds, must also be considered, as it influenced the spatial distribution of related damage.

It could be hypothesised that state aid was probably limited to extreme weather events because they were, by nature, more
localised and exceptional. Providing assistance in the event of other, less extreme but more regular meteorological
phenomena affecting a wider area (Brázdil et al., 2024) would require mobilising a much larger proportion of tax revenue. In
contrast, other systems later emerged to assist populations during periods of frumentary difficulties, such as public granaries

(Brázdil et al., 2025b). Tax relief for extreme weather, from a spatial perspective, was probably easier for the state to manage
from an accounting point of view, as compensation could be balanced by unaffected regions.

### 5.2 Extreme weather and tax alleviations

#### 5.2.1 National context

This study is the third analysis of historical-climatological orientation from southwestern Bohemia with a closer focus on the

Prácheň Region. While the first paper analysed the effects of weather and climate on fluctuations in grain prices (Brázdil et
al., 2024) and the second examined the use of public granary data to identify poor and good grain harvests in relation to
weather extremes (Brázdil et al., 2025b), our recent investigations concentrated on the analysis of taxation records with
respect to weather events causing direct damage to peasants. Comparing the results obtained from southwestern Bohemia
with those from southern Moravia (the southeastern part of the CR), various differences are apparent despite distinct basic

approaches. While tax relief in Bohemia was investigated exclusively based on summarising documents at the regional (the
Prácheň Region) or governmental (Bohemian *Gubernium*) level, the research approach in Moravia started from damage





reports at the lowest level of individual peasants or communities and extended to the regional level of individual estates or domains. Although the First Moravian Land Registry also allowed tax alleviation in cases of "damage by fire or otherwise" from 1655 CE (Novotný, 1934), taxation records related to weather damage became more frequent only from the second half of the 18th century (see, e.g., Brázdil et al., 2012, 2014, 2016; Dolák et al., 2013), similarly to the situation in southwestern Bohemia (see Fig. 8a).

In evaluating peasants' tax relief related to weather damage, the corresponding effects on their livelihood cannot be omitted. For example, Dolák et al. (2015) reported – based on the same type of data – direct negative impacts in the form of poor or failed yields, depletion of livestock, and damage to fields and meadows, as well as losses of property, supplies, and farming equipment. A poor or failed harvest often required borrowing grain for sowing from public granaries, resulting in indebtedness and the obligation to return the grain after the following year's harvest. For instance, in the Horažďovice domain, after wet conditions with hailstorms in 1821 and drought in 1822, peasants failed to return any owed grain to the granary and requested a postponement of repayment until after the 1823 harvest (AS15). From a long-term perspective, a lack of income, debt, impoverishment, reduction in livestock, and deterioration in field fertility could have serious negative impacts on peasants' farming and livelihood (Dolák et al., 2015).

The continuous payment ability of serfs played a key role for the government, as income from land tax formed a fundamental pillar of state revenues in the Habsburg Empire for much of the period analysed, before it was later replaced by indirect taxes (Pekař, 1932; Dickson, 1987). For this reason, the state administration sought ways to ensure the payment capability of serfs. Tax alleviations were thus aligned with reforms influenced by cameralist thinking during the Enlightenment period.

### 5.2.2 International context

Tax support for peasants affected by yield disasters occurred elsewhere in Europe too. One such case occurred in France during the reign of King Louis XI, in response to the famine that affected the Kingdom of France (as well as much of Western Europe) in 1481–1483 CE. This, however, was an exceptional measure (Le Roy Ladurie, 2004). A decree issued in 1639 CE by King Louis XIII granted a permanent tax exemption to the Dauphiné province, located between the Rhône River and the High Alps, to compensate for agricultural disasters affecting this mountainous region (Favier, 2011). From the mid-17th century, this system gradually became standardised throughout the kingdom and continued until the early 18th century (Favier, 2007; Brunier and Krauberger, 2011). Huhtamaa et al. (2022) identified the obligation to pay taxes to the Crown (the Swedish Kingdom) as the main economic burden for peasants in Ostrobothnia (present-day Finland) during the 17th century. While analysing three major volcanic eruptions that led to significant summer cooling and poor grain harvests, they found tax debt records to be a valuable dataset for documenting the varying socioeconomic consequences of those events.

Although the results of this study, as well as other papers concerning Bohemia (Matušíková, 1999) and particularly South Moravia (Brázdil et al., 2003, 2006, 2012, 2014, 2016; Zahradníček, 2006; Chromá, 2011; Dolák et al., 2013, 2015; Dolák, 2016), have demonstrated the great potential of taxation records for the reconstruction of past hydrometeorological events



and their impacts on agriculture, their use in other European countries has remained limited and differently oriented
       compared to the above-cited Czech studies and this research. For example, Grove and Battagel (1983) used data from
       general tax commissions to characterise climatic impacts on the economy of the Sunnfjord region (western Norway) during
       the 17th–18th centuries. A marked increase in the number of petitions for tax relief between 1667 and 1723 CE reflected a
       deterioration of economic conditions resulting from glacier advances. These impacts were particularly evident at high-

altitude farms affected by landslides and avalanches, and at lower-altitude farms suffering from floods. Gjerde et al. (2023),
       analysing the Little Ice Age advance of the Nigardsbreen glacier (western Norway), which culminated in 1748 CE, used a
       novel dataset of local tax load that directly reflected glacial impacts on farming productivity when cross-checked with other
       sources. The potential of tax payment data and tithes for reconstructing wheat and barley yields – and using these to infer
       precipitation levels – was demonstrated for the Canary Islands for the period 1595–1836 CE by García et al. (2003). Związek

et al. (2022) used tax registers and tax exemption data in Poland to document the economic impacts of droughts during
       1531–1540 CE, a period considered by Brázdil et al. (2020) as potentially the driest summer decade in Central Europe over
       the past 500 years. It is remarkable to note that all states across Europe experimented with similar systems to alleviate the
       suffering of populations facing the consequences of extreme weather. This development went hand-in-hand with the
       formation of modern states and reflected the growing perception – especially by the 18th century – that weather-related

crises constituted a form of 'political failure', compelling authorities to take action (Pfister and Wanner, 2021).

       Concerning Asia, a similar system of tax reduction or exemption as in the Czech Lands existed much earlier in Korea during
       the Koryŏ Dynasty (936–1392 CE). As detailed by Oh (2022), three basic types of taxes allowed for reductions or
       exemptions in cases of crop and field damage caused by natural disasters such as floods, droughts, or frosts, as well as pests,
       insects, diseases, or wars. In China, temporary tax exemptions and postponements were employed by most dynasties to

support regions affected by harvest failures. This approach was complemented during the Qing Dynasty (1644–1911 CE) by
       the establishment of state granaries to prevent food crises. Shiue (2005), in a detailed account of famine relief, also presented
       the number of disaster reports submitted by 18 provinces seeking tax relief between 1644 and 1820 CE. About 70 % of these
       reports concerned drought and floods, followed by hail, locusts, typhoons, snowstorms, earthquakes, thunder, and epidemics.
       An analysis of fiscal balance in China in relation to climate change from 220 BC to 1910 CE considered taxes only

marginally (Wei et al., 2014). In Japan, during the Tokugawa (Edo) Shogunate (1603–1868 CE), peasants could request tax
       reductions if crop yields in a given year fell below official expectations (Hayami et al., 2004).

       Governmental support for peasants and farmers affected by weather damage, allowing for a subsequent reduction in their tax
       burden, continued in the Czech Lands until the 1940s. During the 19th century however, farmers began to assume some
       personal responsibility for preventing losses due to harvest damage through the emergence of newly established insurance

companies (see, e.g., Vause, 2023 for France). For example, insurance was mentioned in connection with a flash flood in
       Zbynice in the Prácheň Region on 7 July 1854 (Anonymous, 2025): "[The farmers] *Šafára and Tesárek were already at that
       time insured against hailstorms based on the recommendation of innkeeper Votruba from Sušice [...]. And it was worth it. A
       farmer from Tůma paid an insurance of 10 guldens and received compensation of 400 guldens. He bought rye for sowing*



*and the next year he had* [a good] *harvest, while others left the greater part of their fields fallow due to lack of seed.*"

However, broader hailstorm insurance – generally considered highly hazardous – did not become widespread until the final decades of the 19th century, as seen in regions such as Bavaria (Schmitt-Lermann, 1984), the Czech Lands (Marvan, 1989), and Switzerland (Mauelshagen, 2011).

## 6 Conclusions

Results of the analysis of taxation records as a source of data for historical climatology, using the example of the Prácheň

Region in southwestern Bohemia during the 1655–1827 CE period, can be summarised as follows:

(i) Weather damage to peasants within the valid taxation system was a reason for tax relief. Documents surviving from the corresponding administrative process illustrate its different stages and can be used to study related weather events and their impacts on peasants.

(ii) The use of summarised taxation data at the regional and governmental administrative levels allowed the detection of the

most significant weather events affecting agriculture. However, as a typical institutional source, tax relief data suffer from various uncertainties that must be taken into account in the analysis and interpretation of results.

(iii) Tax relief data represent an important source for the study of convective storms, enabling the analysis of significant hailstorms, torrential rains, and lightning strikes, and allowing the quantification of their spatial extent and the magnitude of related damage affecting the lives of peasants. When complemented by records from other documentary sources, they form a

valuable dataset for creating chronologies of past damaging weather events and their impacts.

(iv) Although the use of institutional data in historical-climatological research in Europe is broad and covers various topics such as climate reconstruction, past weather extremes, and human impacts and responses, the analysis of taxation data in this paper demonstrates the great variety and specific features of documentary data that can be used, at least on a regional scale, for new and original studies.



**Appendix A**

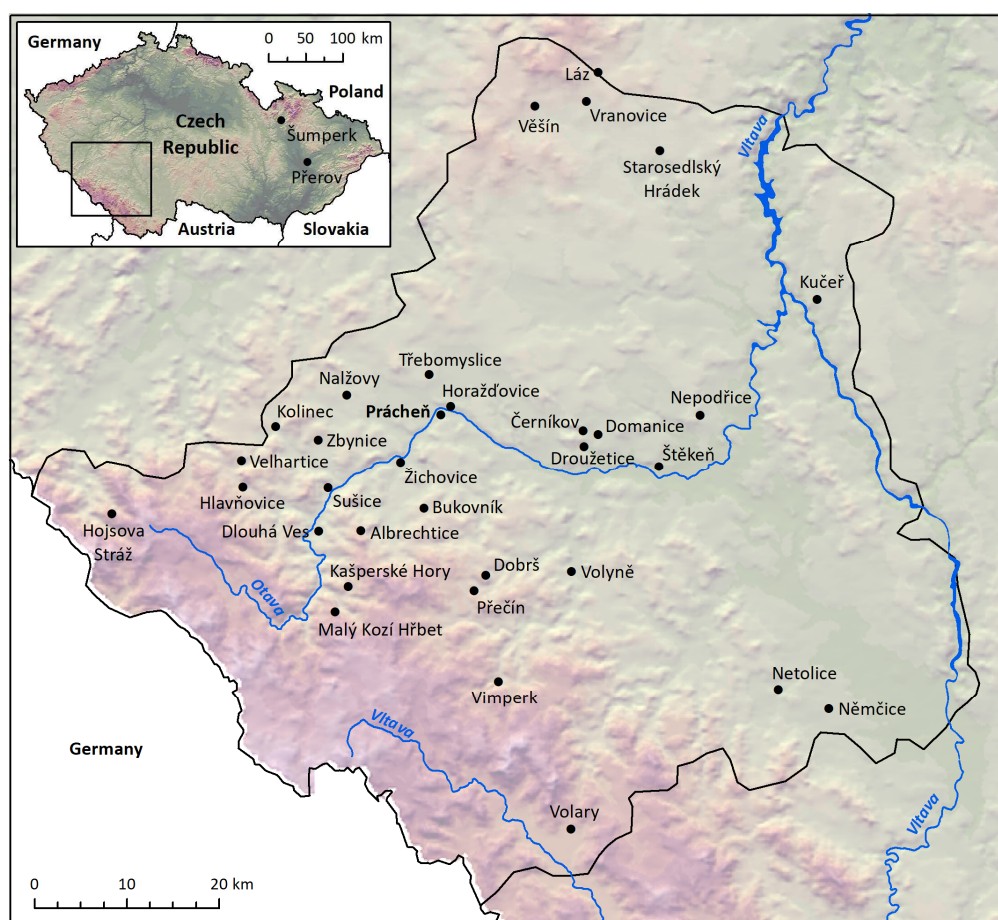

**Figure A1. Location of places over the Czech Republic cited in this paper.**

**Data availability**

The datasets and series used in this article are available from https://doi.org/10.5281/zenodo.15679967 (Brázdil et al., 2025a).

**Author contributions**



RB designed and wrote the paper with contributions from all co-authors. JL collected all taxation data, interpreted them, and contributed to the paper's writing with historical knowledge. KC performed the basic calculations concerning of taxation data as well as weather extremes and finalized all figures. LL contributed with expertise on the taxation systems outside the Czech Lands and to the paper's overall conclusions and approach.

### Acknowledgements

We acknowledge Takehiko Mikami (Tokyo, Japan), Qing Pei (Hong Kong) and Ingar Stene (Oslo, Norway) for discussion of the paper topic and recommendations of references, and Laughton Chandler (Charleston, SC) for English language corrections.

### Financial support

This research was supported by the Johannes Amos Comenius Programme and the Ministry of Education, Youth and Sports of the Czech Republic through the project "AdAgriF Advanced methods of greenhouse gases emission reduction and sequestration in agriculture and forest landscape for climate change mitigation" (CZ.02.01.01/00/22_008/0004635).

### Archival sources

AS1: Národní archiv, fond České gubernium-contributionale, karton 342.

AS2: Národní archiv, fond České gubernium-contributionale, karton 344.

AS3: Národní archiv, fond České gubernium-contributionale, kartony 923–928.

AS4: Národní archiv, fond České gubernium-contributionale, karton 1358.

AS5: Národní archiv, fond České gubernium-contributionale, karton 1922.

AS6: Národní archiv, fond České gubernium-contributionale, karton 2017.

AS7: Národní archiv, fond České gubernium-contributionale, sign. F 1/12, karton 97.

AS8: Národní archiv, fond Nová manipulace, sign. S 3/4, karton 809.

AS9: Národní archiv, fond Nová manipulace, sign. S 3/4, kartony 809–812.

AS10: Národní archiv, fond Nová manipulace, sign. S 3/4, karton 812.

AS11: Národní archiv, fond Repartiční seznamy, kniha č. 157.

AS12: Národní archiv, fond Repartiční seznamy, kniha č. 196.

AS13: Státní oblastní archiv Třeboň, fond Krajský úřad Prácheň, karton 120.

AS14: Státní oblastní archiv Třeboň, fond Krajský úřad Prácheň, karton 120–124.

AS15: Státní oblastní archiv Třeboň, fond Krajský úřad Prácheň, karton 641.

AS16: Státní oblastní archiv Třeboň, fond Krajský úřad Prácheň, karton 641 and 656.

AS17: Státní okresní archiv Klatovy, fond Archiv města Sušice, inv. č. 945, karton 40.

AS18: Státní okresní archiv Klatovy, fond Farní škola Bukovník, inv. č. 25, sign. K 25, Pamětní kniha 1815–1869



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
