# Peer review of "The potential of taxation records as a data source for historical climatology"

_EGUsphere, 2025_

## Author Comment (AC1)

This manuscript evaluates historical records of tax relief to reconstruct weather events and their impacts, using a region of the Czech lands during the 18th century as a case study. As discussed in the manuscript (section 5.2), similar records have been evaluated for other parts of the world. However, this study represents their most detailed and consistent application in European historical climatology. The article therefore makes a substantial contribution to both the climate history of a particular regional and the methods of historical climatology. It is well written, clearly organized, and nearly ready for publication in its current form.

RESPONSE: We would like to thank the reviewer 1 for generally positive evaluation of our study as well as several comments, which we are trying to respond below.

I suggest only that the authors address the following minor issues:

The term "rustical" (line 103 and elsewhere) is not familiar. Please explain it or use "rural".

RESPONSE: On line 102 we mentioned "rustical land (i.e., land held by peasants)". From this explanation follows that "rustical" was used for land in holding by peasants to distinguish it from "dominical", i.e. land belonging to nobility.

Based on the reconstruction in figure 10b, it seems that the frequency of reported hailstorms fell after 1780, while the frequency of other damaging events increased. Should we interpret this as an actual decrease in the frequency of hailstorms? Or is this apparent change more likely to reflect changes in the way that damage was assessed and compensated?

RESPONSE: With respect to great spatiotemporal incompleteness of data and its uncertainties we have to be very careful to formulate any statements concerning of trends in frequency of presented damaging events. We can only describe what we see in Fig. 10b (years or periods of higher or lower frequencies/records), but without any speculation concerning increasing or decreasing tendencies in phenomena recorded. For example, increase in "other damaging events" after 1780 can reflect to the fact, that from 1775 CE it was also allowed to ask for tax relief in case of water damage and there started to appear tax relief for events with both hailstorm and water damage, treated in Fig. 10b as separate events. The higher available number of surviving documentary sources from the 17th to the 19th century also favour an increase in recorded damaging events.

In section 5.1 or 5.2.1, I would like to see a little discussion of how these records might be compared or combined with other sources to help reconstruct past weather and its impacts. After all, we do not know how consistent these sources were. That is, when we have a report of damage then we can probably infer that some damage really occurred. However, when we don't have a report of damage, we don't necessarily know what that means. Maybe there wasn't any extreme weather or damage, or maybe we just don't have a record of damage and tax relief. Do we have any other descriptions or proxies of extreme weather (e.g. hailstorms) in this region and period that we could use to test the consistency of these damage reports? Could we use these damage reports to help understand changing exposure or vulnerability to extreme weather that we have reconstructed from other sources? And how does the frequency of hailstorms and damage in this period compare to the frequency of modern hailstorm in the region?

RESPONSE: The reviewer is fully true expressing doubts about presented data ("After all, we do not know how consistent these sources were. That is, when we have a report of damage then we can probably infer that some damage really occurred. However, when we don't have a report of damage, we don't necessarily know what that means. Maybe there wasn't any extreme weather or damage, or maybe we just don't have a record of damage and tax relief."). We expressed it in Sect. 5.1 Data uncertainty, which follows particularly from involving of

different regulations for tax relief requests (Sect. 2.2) and spatiotemporal incompleteness and details of tax records. Consistency of damage records from tax relief can be tested only from other documentary sources (e.g., chronicles, public granaries), that partly agreed with events covered by tax records and which allowed us to extend tax relief data by other 148 events (see Fig. 10b). General incompleteness of the used data does not allow to characterise "changing exposure or vulnerability to extreme weather" as mentioned by the reviewer. Comparison of "the frequency of hailstorms and damage in this period to the frequency of modern hailstorm in the region" is extremely difficult because of no systematic data concerning damage. Based on Czech Hydrometeorological Institute database, data from our studied region are rather sparse, with frequent changes of available stations and mostly short period of their observations, from which is extremely difficult to create any valuable series. Trying to take above reviewer comments in account, we added the new second paragraph in Sect. 5.1 Data uncertainty as follows:

"Reported uncertainties have to be taken into account in evaluation of the results obtained. It concerns, for example, interpretation of frequencies of damaging weather events in summarising Fig. 10, for which any conclusions about longer increasing or decreasing tendencies in frequency of presented events would be very speculative. Moreover, rather local occurrence of such phenomena like hailstorms or torrential rains complicate their observation even in the recent network of meteorological stations of the Czech Hydrometeorological Institute (CHMI), moreover do not recording any damage. Frequent changes in stations and periods of their observations are reflected in missing studies dealing with spatiotemporal variability in any such areas as represented by Prácheň Region in our study, except of existing systematic hailstorm analyses from the eastern part of the Czech Republic (Chromá et al., 2005; Brázdil et al., 2016)."

New reference:

Chromá, K., Brázdil, R., and Tolasz, R.: Spatio-temporal variability of hailstorms for Moravia and Silesia in the summer half-year of the period 1961–2000, Meteorol. Čas., 8, 65–74, 2005.

In section 5.2: Similar tax support was also used in the event of natural disasters in the Ottoman Empire. See Sam White, *Climate of Rebellion* (Oxford UP, 2011), p.79-85.

RESPONSE: The following sentence was complemented at the end of second paragraph in Sect. 5.2.2 as follows:

"Tax support of farmers has been also used in the case of damaging hydrometeorological phe⬜ome⬜a i⬜the Ottoma⬜Empire (Ursi⬜us, 1999; White, 2011)."

New references:

Ursinus, M.: Natural disasters and Tevzi: Local tax systems of the post-classical era in response to flooding, hail, and thunder, in: Natural Disasters in the Ottoman Empire, edited by: Zachariadou, E., Crete University Press, Heraklion, 281 pp., ISBN 978-960-524-092-9, 1999.

White, S.: The Climate of Rebellion in the Early Modern Ottoman Empire, Cambridge University Press, New York, USA, 376 pp., ISBN 9780511844058, 2011.

---

## Author Comment (AC2)

First, I would like to thank the editors of *Climate of the Past* for asking me to review this very interesting article, which uses previously unpublished documentary archives as a source for historical climatology in Europe. The authors are highly qualified climate historians, and the subject of the article is very much in line with some of the themes explored by the journal. This article is the third analysis of historical-climatological orientation from southwestern Bohemia (Czech Republic) published by the same authors in CfP in recent years.

RESPONSE: We would like to thank Nicolas Maughan for the reviewing of our study with many comments, which we are trying to respond below.

**General comments**:
- Paragraph "2.4 Climatic data": this section could perhaps be expanded with a few sentences to provide a more detailed description of the climatic context during the period studied in Central Europe. The same comment applies to Figure 10 and text from line 353 to 359.

In this section, the authors could have used the powerful new tool called ClimeApp (https://mode-ra.unibe.ch/climeapp/) to produce an additional figure, map or time series to show precipitation or temperature anomalies during this period. It is a web-based tool for processing paleoclimate data, presenting temperature, precipitation, and pressure reconstructions from 1422 to 2008 CE (based on ModE-RA & ModE-RAclim global climate reanalysis). This tool is very easy to use.

RESPONSE: Of course, we know about ModE-RA reanalysis (Valler et al., 2024, https://doi.org/10.1038/s41597-023-02733-8), presenting among others gridded reconstructions of temperature and precipitation over Europe. But having corresponding temperature and precipitation reconstructions for the Czech Lands, presented in Fig. 10, we do not see as useful to use Valler et al. paleoclimate reconstruction for Central Europe, because it uses among basic series also our Czech temperature and precipitation reconstructions by Dobrovolný et al. (2010, 2015), i.e. it represents not independent data source.

- The use of administrative documentation connected with requests for tax relief from peasants in southwestern Bohemia during the 17th–19th centuries (to identify extreme weather events and establish a chronology) is a very good idea. Indeed, these documentary archives can serve as an important source of data for historical climatology ate regional level.

RESPONSE: We agree with the reviewers about importance of tax relief data for historical climatology on the regional level.

- The method used (part 3) enabled a total of 2,134 records to be collected from the period 1655 to 1827 CE and, even after eliminating data relating to fires (unrelated to weather), this remains a substantial dataset (posted online by the authors). The 1,107 individual taxation records related to weather damage (this dataset was further complemented by another dataset from documentary sources) were classified into four categories and analyzed individually, followed by a comprehensive analysis of damaging weather events over the period 1655–1827 CE. This methodology is rigorous and allows the data to be presented in a clear and highly visual manner.

RESPONSE: Many thanks for the positive evaluation of the used methodology.

- It is a real shame that due to the lack of data during certain periods, 1708–1747 and 1806–1813 CE (5.1 Data uncertainty, lines 363 to 365)  *"This also applies to institutional evidence represented in this study by taxation records, which is particularly reflected in data gaps between 1708 and 1747 CE and further in the years 1806–1813 CE"*, the consequences of

extreme climatic events such as the famous winters of 1709 and 1740 could not be analyzed in the Prácheň Region.

RESPONSE: The two famous severe winters 1708/09 and 1739/40 CE occurred in the period not covered by taxation data. Moreover, potential impact of "freezing of winter crops" caused by hard frosts or long lying snow, as mentioned e.g. for 1751 or 1758 CE in Sect. 4.1.4, were not among meteorological phenomena, with which was possible to argue for tax alleviation. But there exist other documentary sources (particularly chronicles) from the studied region or other locations in Bohemia, in which above two extreme winters were reported.

- Paragraph 5.1 concerning data uncertainty in the use of documentary evidence in historical-climatological research is important and very welcome, as it highlights the obstacles to the use of documentary archives. The description of these problems associated with the use of documentary archives, which are well known to historians but less so to other academic disciplines, is important because it allows the advantages and disadvantages of the extracted data to be identified. This facilitates communication between history, geography and geosciences in the broad sense, making it possible to correct and supplement incomplete data (such as early meteorological records) with the help of other disciplines and to build solid interdisciplinary studies in climate history.

RESPONSE: Many thanks for the positive evaluation of data uncertainty.

- The article is well structured, organized in a traditional manner around six sections. It is written in good quality English. The figures and maps are clear and very well presented. The bibliography is exhaustive and very recent, drawing on the most relevant works on the subject, with many of the articles cited relating to similar case studies from other countries around the world.

RESPONSE: Many thanks for the positive evaluation of our paper.

**Specific questions about the documentary archives used**:

- In the abstract, and then in the text, the same information is given about the type of event, the damages, taken into account in the registers and when they began to be taken into account:

(abstract, line 12-14) : "*based on the first land registry system, only hailstorm damage to crops and fires qualified peasants for tax relief from 1655 CE, while the subsequent land registry system from 1748 CE extended this to include water damage from 1775 CE.*"

(2.2 Taxation system and data, line 119-120): "*From 1775 CE, the reporting of key events for tax alleviation was extended from fire and hailstorm to also include water damage.*"

If I understand correctly, a new registry system was created in 1748 but did not begin recording water damages until 1775. Do the authors know why it was decided to record this damage only from 1775 onwards and not from 1748? Perhaps because a more turbulent climate period (increased rainfall) in southwestern Bohemia would have considerably increased damage to crops and made it necessary to take this into account in order to help the population? This is not specified and would benefit from some clarification for readers.

RESPONSE: You are true, that a new registry system involved in 1748 CE took in account again only hailstorms and fires for tax relief and water damage as a reason for such relief was involved as far as from 1775 CE. Although some attempts for extension of considered hydrometeorological events (like windstorms, severe frosts, floods) appeared several times from the beginning of the 18th century, finally they were not accepted. We may only speculate that it could have been limited by available volume of financial sources (188,000 guldens) for compensation of damage. Although in the 1750s water damage was identified among very frequent cases, not any reasons for its later involving in Bohemia from 1775 CE were mentioned.

- The information provided about the damage assessment process (2.2 Taxation system and data, lines 84–91, and figure 2) is particularly interesting because it presents a mechanism that was almost identical to one that existed during the same period in the 'States of Provence' (South-eastern France) in the 17th and 18th centuries. After a natural disaster, rural communities (peasants) would submit requests, known as "requests for assistance from rural communities", to the central administration of Provence.

These requests for tax relief were assessed on the same principle as that described in the Czech Republic and also represent a very rich source of data (a huge corpus of documentary archives) for climate history in South-eastern France, which is currently being studied. The authors accurately refer to these tax exemptions, sometimes temporary, for certain regions of France ("5.2.2 International context. lines 417 to 421), but their scope was much greater and the requests were very regular, especially in the mountainous areas of the south-east of the country (southern part of the French Alps), which are prone to extreme climatic events and intense soil erosion (it is a pity that French data are never taken into account in European studies...). Overall, this section is very well constructed and provides many examples of systems of tax reduction or exemption that have been used around the world to infer precipitation levels or document the economic impacts of droughts.

RESPONSE: Thanks for your comments concerning of similar data from South-eastern France. The fact mentioned by the reviewer, "that French data are never taken into account in European studies", can be perhaps related to publishing such papers in French language, when its poor knowledge among scientists could be an obstacle for the use and citing of such papers.

- I assume that there are two main types of documents (according to Figure 2): the request from the community affected by damage, the "report", and a 'control' document specifically from the regional administration's inspection, in which it is also possible to find information on the actual extent of the damage?

In the case of South-eastern France, where a similar process existed, the information in these "inspection" documents is very interesting because analysing it allows us to identify certain communities that had a bad habit of regularly overestimating the damage or even requesting tax exemptions for disasters that had not occurred... (like false insurance claims today...). These communities eventually became known to the authorities... Of course, this was not the case for the majority of claims made after a disaster.

Thus, it is possible to eliminate or downplay some specific weather events and thus obtain a more reliable data set. Could the authors have carried out similar work with the documents available in the Prácheň Region?

RESPONSE: As follows from our study related to taxation records in South Moravia (south-eastern part of the Czech Lands), reports of damage from individual communities sent to the Regional office were checked in situ by office authorities. In this evidence appeared sometimes cases, when an original request on tax relief from community was significantly reduced (see e.g. Brázdil and Valášek, 2003), but we did not find that it concerned systematically only one or any particular communities. Concerning of Bohemia, reports of caused damage were submitted by landlord's office to the Regional office and its authority checked it in situ and determined a measure of damage. This system eliminated potential overestimation of the damage caused. Because our recent study used such data on the level of the Regional office or *Gubernium*, not directly from individual communities, we could not identify communities "regularly overestimating the damage or even requesting tax exemptions for disasters that had not occurred" as in South-eastern France.

- The periods of the major volcanic eruptions of Laki (1783) and Tambora (1815) are covered by the available archives, but they are not mentioned at all in the article (just as an example in Sweden). Is this because it is not possible to highlight their local climatic consequences based on the information provided by the documents analysed?
RESPONSE: Climatic and socioeconomic consequences of Laki and Tambora eruptions were analyzed by Brázdil et al. (2017): Climatic and other responses to the Lakagígar 1783 and Tambora 1815 volcanic eruptions in the Czech Lands, https://doi.org/10.37040/geografie2017122020147. Concerning of damage records in the Prácheň Region from the years of the eruption and several subsequent years, there were no direct indications of potential volcanic effects. It was a reason, that both eruptions were not particularly mentioned in our recent study.

- It is a pleasure to read a well-structured article such as this one. The methodology used could certainly serve as a model applicable in other countries in Europe (such as France) or Asia where similar documentary archives exist. I therefore recommend its publication in CfP after a few minor revisions.
RESPONSE: Thank you for your positive evaluation.

**Comments for minor revisions**:
-    Section "2.4 Climatic data" needs to be expanded slightly.
RESPONSE: Please see our response concerning of the use of ModE-RA reanalysis (Valler et al., 2024) above. We believe, that the use of temperature and precipitation reconstructions for the Czech Lands (Dobrovolný et al., 2010, 2015) gives a valuable information expected by the reviewer.

-    A few sentences should be added to the conclusion "perspectives", for example regarding similar documents available in other regions of the Czech Republic that could be studied for historical climatology in the future.
RESPONSE: Accepted, the following paragraph was complemented to Conclusion as a new point (iv) in the following form:
"(iv) While until now a systematic research of taxation records for the study of damaging hydrometeorological extremes in the Czech Lands concerned only South Moravia, the analysis of the same datasets for the Prácheň Region shows importance of these data also for Bohemia. Although the work with taxation data represents extremely time-consuming work, it opens their enormous future potential for significant extension of knowledge related to hydrometeorological extremes and their socioeconomic impacts during 17th–19th centuries over the whole Czech Republic."

- It would be useful to add some geographical details to the maps in Figure 9 and in the legend, such as the name of the Prácheň Region.
RESPONSE: Accepted, please see the new version of Fig. 9 below.

[Figure]

Figure 9. Spatial extent of four outstanding damaging weather events with the highest number of affected municipalities: (a) 15 June 1758, (b) 13 July 1763, (c) 21 June 1765, (d) 31 July 1779; 1 – hail damage, 2 – hail and water torrent damage.